# Consistent Labeling Across Group Assignments: Variance Reduction in Conditional Average Treatment Effect Estimation

## Abstract

Numerous algorithms have been developed for Conditional Average Treatment Effect (CATE) estimation. In this paper, we first highlight an overlooked issue in CATE estimation: many algorithms exhibit inconsistent learning behavior for the same instance across different group assignments. We introduce a metric to quantify and visualize this inconsistency. Next, we present a theoretical analysis showing that this inconsistency indeed contributes to higher test errors and cannot be resolved through conventional machine learning techniques. To address this problem, we propose a general method called **Consistent Labeling Across Group Assignments** (CLAGA), which eliminates the inconsistency and is applicable to any existing CATE estimation algorithm. Experiments on both synthetic and real-world datasets demonstrate significant performance improvements with CLAGA.

## 1 Introduction

In recent years, personalized decision-making has become increasingly prevalent, spanning areas such as medicine (Bica et al., 2021; Foster et al., 2011; Jaskowski & Jaroszewicz, 2012; Li et al., 2024), marketing (Gubela et al., 2019; Machluf et al., 2024; Wang et al., 2025), and education (Zhang, 2025; Luke Miratrix, 2025). A critical component of personalized decision-making is understanding how different individuals or subgroups respond differently to interventions. This is where the Conditional Average Treatment Effect (CATE) plays a central role. CATE estimates the expected effect of a treatment conditional on certain characteristics of the individual or subgroup, allowing for targeted and effective interventions.

Estimating CATE accurately from real-world data, however, is a challenging task. Existing CATE estimation methods typically follow the potential outcomes framework (Rubin, 1974; 2005), which quantifies treatment effects by comparing the outcomes an individual would have experienced if they had received the treatment versus if they had not. The main challenge in estimating CATE lies in the fact that we can only observe the outcome under one condition—either with or without the treatment—making it impossible to directly measure the treatment effect for an individual.

To address this challenge, machine learning approaches have adopted various strategies (Zhang et al., 2021a; Gutierrez & Gérardy, 2017), with the goal of generating learning targets that can (a) facilitate supervised learning for training estimators, and (b) ensure unbiased estimation. Among the early methods, both the single-model approach (Lo, 2002) and the two-model approach (Hansotia & Rukstales, 2002) have remained competitive over the years. More recently, advanced techniques such as the X-learner (Künzel et al., 2019), R-learner (Nie & Wager, 2021), and Doubly-Robust learner (Kennedy, 2020) have been proposed. In parallel, neural network–based models such as TarNet and CFRNet (Shalit et al., 2017) and DragonNet (Shi et al., 2019) have further extended CATE estimation to deep learning frameworks. These modern methods all follow the principle of transforming the CATE estimation problem into one or more supervised learning tasks.

Different from prior works, which primarily focus on unbiased estimation, this paper identifies an overlooked phenomenon in CATE estimation: inconsistent learning across treatment groups. We show that this inconsistency directly contributes to estimation error, a source of inaccuracy not

previously recognized in the literature. Building on this insight, our work emphasizes both the variance and statistical efficiency of CATE estimators. Our contributions are threefold:

1. **Identification of Inconsistent Learning across Group Assignments**: We reveal the problem of inconsistent learning behavior in CATE estimation algorithms, where the prediction for a given training instance differs significantly depending on whether it is assigned to the treatment group or the control group. This inconsistency persists even with large datasets, up to 1 million instances. To quantify this effect, we introduce a metric, termed the *discrepancy ratio*.

2. **Theoretical Framework for Error Decomposition**: We propose a theoretical framework that decomposes CATE estimation errors into distinct components, with a subset identified as being linked to group assignment inconsistencies. We show that these inconsistencies indeed contribute to higher estimation errors and are inherent to algorithm design, making them resistant to traditional machine learning solutions such as hyperparameter tuning or model selection.

3. **Proposed Solution for Consistent Labeling**: We introduce a method called *consistent labeling across group assignments* (CLAGA), which is applicable to any off-the-shelf CATE algorithm. By applying this method, the error term related to inconsistent learning across group assignments is effectively eliminated. Our experimental results, conducted on both synthetic and real-world datasets, demonstrate significant performance improvements, supporting the validity and practical benefits of our proposed method.

## 2 PROBLEM FORMULATION

In this paper, we follow the potential outcomes framework proposed by Rubin (1974; 2005). In this framework, each instance is represented by a feature vector $X \in \mathcal{X}$. Each instance $X_i$ is assigned a treatment indicator $W_i \in \{0, 1\}$, which indicates whether the treatment is received. Additionally, each instance has two potential outcomes, $Y_i^{(0)}$ and $Y_i^{(1)}$, which represent the outcome if the treatment is not received ($W_i = 0$) and if the treatment is received ($W_i = 1$), respectively.

We adopt the standard assumptions of "consistency", "unconfoundedness", and "positivity" within the potential outcomes framework (Rubin, 2005), which are defined as follows:

- **Consistency**: The observed outcome $Y$ is equal to the potential outcome corresponding to the received treatment assignment:

$$Y = \begin{cases} Y^{(1)} & \text{if } W = 1, \\ Y^{(0)} & \text{if } W = 0. \end{cases}$$

- **Unconfoundedness**: Conditional on covariates $X$, the treatment assignment is independent of the potential outcomes:

$$(Y^{(0)}, Y^{(1)}) \perp\!\!\!\perp W \mid X$$

- **Positivity (Overlap)**: Each unit has a non-zero probability of receiving either treatment:

$$0 < \Pr(W = 1 \mid X = x) < 1 \quad \text{for all } x \in \mathcal{X}.$$

The conditional potential outcome functions, $\mu_0(x)$ and $\mu_1(x)$, denote the expected outcomes under the two group assignments. Specifically, $\mu_0(x) = \mathbb{E}[Y^{(0)} \mid X = x]$ and $\mu_1(x) = \mathbb{E}[Y^{(1)} \mid X = x]$. The Conditional Average Treatment Effect (CATE), $\tau(x)$, is defined as the difference between these two potential outcome functions:

$$\tau(x) = \mu_1(x) - \mu_0(x). \tag{1}$$

In a typical CATE estimation scenario, we are given a dataset $\mathcal{D} = \{(X_i, W_i, Y_i)\}_{i=1}^n$, where each unit $i$ is drawn i.i.d. from a joint distribution over $(X, W, Y)$, and the observed outcome satisfies $Y_i = Y_i^{(W_i)}$. Since only one of the two potential outcomes $(Y_i^{(0)}, Y_i^{(1)})$ can be observed for each individual, direct supervision of the treatment effect function $\tau(x)$ using standard supervised learning methods is infeasible.

Note that $\tau(x) = \mu_1(x) - \mu_0(x)$ is a deterministic function of the covariates $x$; it is not a random variable and does not depend on treatment assignment, i.e., it is assignment-invariant. The central challenge of CATE estimation lies in inferring this function from only partial observations of the potential outcomes.

Although we list the standard assumptions of the potential outcomes framework for completeness, our analysis and proposed method do not rely on the unconfoundedness assumption. In particular, our method applies to both randomized controlled trials (RCTs), where unconfoundedness holds by design, and observational settings, where the assumption may not be plausible. We only require that the dataset $(X_i, W_i, Y_i)$ is drawn i.i.d. from a joint distribution, and that the observed outcome satisfies $Y_i = Y_i^{(W_i)}$.

While different algorithms employ a variety of strategies to address this challenge, in Section 4, we demonstrate that most of these strategies lead to inconsistent learning across group assignments. In Section 5, we further show that such inconsistencies contribute to higher CATE estimation errors.

## 3 RELATED WORK

### 3.1 ALGORITHMS FOR CATE ESTIMATION

CATE estimation is challenging due to the fact that we only observe one of the two potential outcomes for each instance. To address this, various strategies have been developed to enable models to learn from the incomplete information. In this paper, we consider state-of-the-art and widely-used CATE estimation algorithms, including the Single-model approach (Lo, 2002), the Two-model approach (Hansotia & Rukstales, 2002; Radcliffe, 2007), the X-learner (Künzel et al., 2019), the R-learner (Nie & Wager, 2021), and the Doubly-Robust (DR) learner (Kennedy, 2020). We also include neural network-based methods such as TarNet and CFRNet (Shalit et al., 2017), and DragonNet (Shi et al., 2019).

Beyond algorithm design, recent work has proposed improvements to CATE estimation from different angles. For example, Johansson et al. (2016) introduced regularization to promote similarity in representations between the treatment and control groups. Yao et al. (2018) proposed preserving local similarity in the feature space. Zhang et al. (2021b) developed a method to disentangle covariate factors to improve CATE estimation.

### 3.2 ANALYSIS OF CATE ALGORITHMS

Early studies focused on comparing and selecting among CATE estimation algorithms, which remains an ongoing challenge, as no single method dominates across all data scenarios (Dorie et al., 2019). Several studies have constructed benchmarks using synthetic and real-world datasets to assess algorithmic performance under diverse conditions (Wendling et al., 2018; Knaus, 2022; Schuler et al., 2017). Other works investigate model selection and performance evaluation criteria, including recent proposals for theoretically grounded metrics (Alaa & Van Der Schaar, 2019; Curth & van der Schaar, 2021).

Recent studies further underscore the need for comprehensive analyses of CATE algorithms. For instance, Yu & Sun (2024) evaluate 16 modern estimators on more than 43,000 dataset variants and find that many methods fail to outperform trivial baselines in terms of MSE on real-world data, revealing inconsistency in performance across treatment groups and domain settings. Huang et al. (2024) proposes a distributionally robust metric that is nuisance-free and aims to select estimators performing well under covariate shift and hidden confounding. Machluf et al. (2024) highlight ensemble-based approaches as a promising direction to improve stability and robustness in CATE estimation.

While these studies emphasize aggregate accuracy or bias metrics, less attention has been paid to whether algorithms learn consistent label functions across treatment groups—a focus of our work. Rather than proposing a new CATE estimator, our approach complements existing algorithms by addressing variance-related inconsistencies that arise even in large samples.

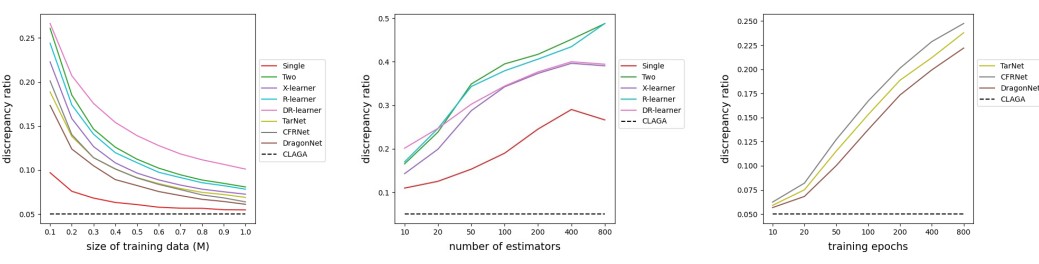

(a) Data size (all algorithms)    (b) Model complexity (tree-based)    (c) Model complexity (NN-based)

Figure 1: (a) The discrepancy ratio decreases as data size increases, suggesting reduced overfitting. (b) Tree-based models with more estimators show higher discrepancy ratios. (c) Neural networks with more training epochs also show increased overfitting risk. In all figures, the black dotted line represents our proposed method, CLAGA, which achieves a consistently lower discrepancy ratio.

## 4    INCONSISTENT LEARNING ACROSS GROUP ASSIGNMENTS

In this section, we introduce and analyze an overlooked phenomenon in CATE estimation, which we term *inconsistent learning across group assignments*. Many CATE estimation algorithms have been developed to address the challenge that the ground truth treatment effect $\tau(x)$ cannot be directly observed. These algorithms often claim to satisfy *pointwise unbiasedness*, meaning that $\mathbb{E}_{M,W}[\hat{\tau}(x)] = \tau(x)$, where the expectation of the estimated treatment effect $\hat{\tau}(x)$ is over both training randomness $M$ and treatment assignment $W$. However, this property alone does not ensure that the learned function behaves consistently when the same individual appears in different treatment groups across runs.

By definition, the true treatment effect $\tau(x) = \mu_1(x) - \mu_0(x)$ is a deterministic function of the covariates $x$ and does not depend on the treatment assignment $W$; it is assignment-invariant. One may therefore reasonably expect a consistent estimator to produce assignment-invariant estimates in expectation:
$$\mathbb{E}[\hat{\tau}(x) \mid X = x, W = 0] = \mathbb{E}[\hat{\tau}(x) \mid X = x, W = 1].$$
In practice, however, most CATE estimators are not guaranteed to inherit this property, as their learned values may differ depending on whether the instance is drawn from the treatment or control group.

Any deviation from the equality suggests that the estimator is implicitly learning different functions depending on group assignment, which contradicts the fundamental definition of an assignment-invariant treatment effect. We return to a more detailed discussion of how this issue relates to CATE estimation error in Section 5.

**Discrepancy Ratio.**    To quantify this inconsistency, we introduce a metric called the *discrepancy ratio*, which measures the extent to which CATE estimators produce significantly different predictions for the same instance depending on its treatment assignment during training.

To compute this metric, we utilize a synthetic dataset, Zenodo (Zhao, 2022) , in which both potential outcomes and the ground-truth $\tau(x)$ are known. We repeatedly train each CATE estimator 30 times on the same dataset, but with randomized treatment assignments for all training instances in each run. For each instance $x$, we collect the predicted treatment effects $\hat{\tau}(x)$ separately for when it was assigned to the treatment group ($W = 1$) versus the control group ($W = 0$).

We then conduct an independent t-test for each instance to determine whether the predictions under the two assignment groups are significantly different. The discrepancy ratio is defined as the proportion of instances where the difference is statistically significant (p-value $< 0.05$).

This metric provides a direct way to measure how often estimators learn assignment-dependent treatment effect functions, thereby violating assignment-invariance and introducing additional variance.

**Empirical Observations.**    Figure 1 shows two key trends of the discrepancy ratio. In Figure 1a, we observe that the ratio decreases as the training data size increases, which aligns with the expectation

that more data mitigates overfitting to group-specific targets. However, even at a scale of 1 million samples, many algorithms still exhibit substantial inconsistency.

Figure 1b and Figure 1c further reveals that as model complexity increases, the discrepancy ratio rises, indicating that complex models are more susceptible to overfitting to assignment-specific noise. This reinforces the intuition that over-parameterized models, when trained on partial observations, can capture spurious patterns that depend on group assignments—despite the fact that the true $\tau(x)$ should remain assignment-invariant.

To address this issue, in Section 6 we will propose a general procedure called *Consistent Labeling Across Group Assignments* (CLAGA), which can be applied to any CATE estimation method. As indicated by the black dotted line in both subfigures, CLAGA inherently makes CATE estimation assignment-invariant and consistently reduces the discrepancy ratio across data scales and model complexities.

## 5 IMPACT OF INCONSISTENT LEARNING

### 5.1 ERROR DECOMPOSITION

In this section, we analyze how inconsistent learning across group assignments impacts the accuracy of CATE estimation. Specifically, we focus on the Precision in Estimation of Heterogeneous Effect (PEHE) (Hill, 2011), which is defined as:

$$\text{PEHE} := \mathbb{E}_x \left[ \mathbb{E}_{M,W,\epsilon} \left[ (\hat{\tau}(x) - \tau(x))^2 \mid X = x \right] \right]. \tag{2}$$

Here, $M$ denotes the randomness introduced by the model training process (e.g., due to different initializations or optimization stochasticity), and $W$ refers to the randomness in treatment group assignment. $\epsilon$ captures the residual variation in the observed outcomes, which may arise either from measurement error or from inherent stochasticity of the outcome distribution (e.g., Bernoulli variability). Specifically, the observed potential outcomes are assumed to follow

$$Y(w) = \mu_w(X) + \epsilon_w,$$

where $\mathbb{E}[\epsilon_w \mid X] = 0$.

For clarity, we omit the explicit dependence on covariates $x$ and conduct the following analysis conditioned on a fixed value of $x$ in the remainder of this section. The extension to the full PEHE over the marginal distribution of $X$ can be recovered by taking the outer expectation over $x$.

To understand how different sources contribute to PEHE, we decompose the squared error $\mathbb{E}_{M,W}[(\hat{\tau} - \tau)^2]$ as follows, with the full derivation and a brief explanation of each error term given in Appendix A.

$$\mathbb{E}_{M,W,\epsilon}[(\hat{\tau} - \tau)^2] = \underbrace{\mathbb{E}_{M,W,\epsilon}[(\hat{\tau} - \tilde{\tau}^{(W)})^2]}_{\text{Model Error}} - \underbrace{2\mathbb{E}_{M,W,\epsilon}[(\tau - \tilde{\tau}^{(W)})(\hat{\tau} - \tilde{\tau}^{(W)})]}_{\text{Model-Target Covariance}}$$

$$+ \underbrace{(1 - \pi)\text{Var}_\epsilon[\tilde{\tau}^{(0)}] + \pi\text{Var}_\epsilon[\tilde{\tau}^{(1)}]}_{\text{Group Assignment Weighted Variance}} + \underbrace{\pi(1 - \pi)(\mathbb{E}_\epsilon[\tilde{\tau}^{(0)}] - \mathbb{E}_\epsilon[\tilde{\tau}^{(1)}])^2}_{\text{Inconsistency Across Group Assignments}} \tag{3}$$

$$+ \underbrace{(\mathbb{E}_{W,\epsilon}[\tau - \tilde{\tau}^{(W)}])^2}_{\text{Bias of Learning Target}},$$

where

- $\tilde{\tau}$ denotes the learning target used by the model for a given instance. In many algorithms, this target is constructed conditional on the observed treatment assignment $W$. Depending on the algorithm design, $\tilde{\tau}$ may take the form of an explicit pseudo-outcome label (e.g., in Single-model approach) or an implicit optimization objective (e.g., in R-learner).
- $\tilde{\tau}^{(0)}$ and $\tilde{\tau}^{(1)}$ represent the learning targets corresponding to the same instance if it were assigned to the control or treatment group, respectively.
- $\pi = \Pr(W = 1|x)$ denotes the treatment assignment probability (with $1 - \pi$ for control).

## 5.2 Insights into the Error Components

We can categorize the error components in Equation 3 into two groups: those that depend on the model training process and those that do not. Specifically, we classify any component involving $\hat{\tau}$ as training-dependent, and those involving only $\tilde{\tau}$ or $\tau$ as independent of model training.

### 5.2.1 Error components unrelated to the training process

The following components are independent of the model training process and arise solely from the construction of the learning target $\tilde{\tau}^{(W)}$, conditioned on treatment assignment $W$:

- $(\mathbb{E}_{W,\epsilon}[\tau - \tilde{\tau}^{(W)}])^2$
- $(1 - \pi) \operatorname{Var}_\epsilon[\tilde{\tau}^{(0)}] + \pi \operatorname{Var}_\epsilon[\tilde{\tau}^{(1)}]$
- $\pi(1 - \pi) (\mathbb{E}_\epsilon[\tilde{\tau}^{(0)}] - \mathbb{E}_\epsilon[\tilde{\tau}^{(1)}])^2$

For convenience, we introduce the following shorthand notations for two of the above terms:

- $\operatorname{WVG}(\tilde{\tau}) \coloneqq (1 - \pi) \operatorname{Var}_\epsilon[\tilde{\tau}^{(0)}] + \pi \operatorname{Var}_\epsilon[\tilde{\tau}^{(1)}]$, which we refer to as the *weighted variance across group assignments*.
- $\operatorname{SDMG}(\tilde{\tau}) \coloneqq (\mathbb{E}_\epsilon[\tilde{\tau}^{(0)}] - \mathbb{E}_\epsilon[\tilde{\tau}^{(1)}])^2$, which we refer to as the *squared difference of mean across group assignments*.

These components depend only on the treatment assignment $W$ and the algorithm's internal design of $\tilde{\tau}$, but not on any specific model training process. Consequently, their magnitudes cannot be mitigated by better model training or hyperparameter tuning—they must be addressed through improved learning target construction.

For instance, $(\mathbb{E}_{W,\epsilon}[\tau - \tilde{\tau}^{(W)}])^2$ vanishes when the learning target $\tilde{\tau}$ is unbiased for $\tau$. Meanwhile, $\operatorname{WVG}(\tilde{\tau})$ increases when the learning targets exhibit higher variance within each group assignment, and $\operatorname{SDMG}(\tilde{\tau})$ increases when their groupwise means diverge.

*Remark.* $\operatorname{WVG}(\tilde{\tau})$ decreases as the variance of $\tilde{\tau}^{(0)}$ and $\tilde{\tau}^{(1)}$ becomes smaller.

*Remark.* $\operatorname{SDMG}(\tilde{\tau}) = 0$ when $\mathbb{E}_\epsilon[\tilde{\tau}^{(0)}] = \mathbb{E}_\epsilon[\tilde{\tau}^{(1)}]$, indicating consistent learning targets across group assignments.

### 5.2.2 Error components related to the training process

The remaining two components involve $\hat{\tau}$, the model prediction, and are therefore sensitive to the randomness introduced by the training process $M$:

- $\mathbb{E}_{M,W,\epsilon}[(\hat{\tau} - \tilde{\tau}^{(W)})^2]$
- $-2 \mathbb{E}_{M,W,\epsilon}[(\tau - \tilde{\tau}^{(W)})(\hat{\tau} - \tilde{\tau}^{(W)})]$

The first term measures how well the model fits the learning target $\tilde{\tau}$, averaged over training and treatment randomness. The second is a covariance term between the learning target's bias and the model's prediction error. It reflects whether the model compensates for the bias of learning target or exacerbates it.

To understand its contribution, we derive an upper bound using the Cauchy–Schwarz inequality:

$$
\begin{aligned}
&\left| -2\mathbb{E}_{M,W,\epsilon}[(\tau - \tilde{\tau}^{(W)})(\hat{\tau} - \tilde{\tau}^{(W)})] \right| \\
&\leq 2\sqrt{\mathbb{E}_{M,W,\epsilon}[(\tau - \tilde{\tau}^{(W)})^2]} \cdot \sqrt{\mathbb{E}_{M,W,\epsilon}[(\hat{\tau} - \tilde{\tau}^{(W)})^2]}.
\end{aligned}
\tag{4}
$$

This shows that the magnitude of this term is upper bounded by the geometric mean of target bias and model error. Assuming models are well-trained with a well-designed learning targets, this term is negligible.

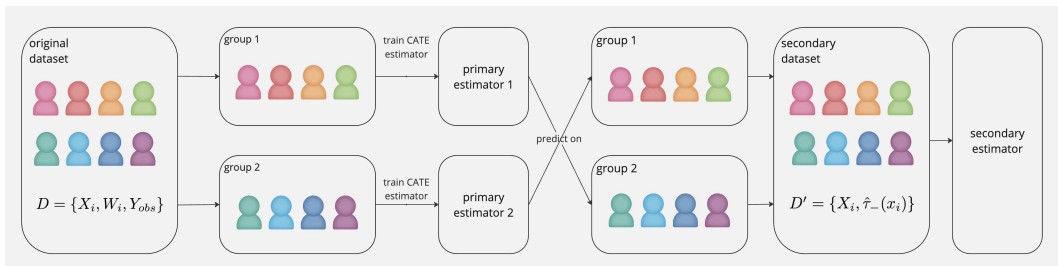

Figure 2: An overview of CLAGA with K=2.

# 6 CONSISTENT LABELING ACROSS GROUP ASSIGNMENTS

## 6.1 MOTIVATION

In Section 5.2, we identified two types of error components. While model training techniques can help reduce the errors related to the training process, our focus here is on improving the inherent errors caused by the algorithm design, as discussed in Section 5.2.1. We propose a general method that can be applied to any CATE estimation algorithm to effectively mitigate these errors.

## 6.2 METHOD

We introduce a method called *consistent labeling across group assignments* (CLAGA), which aims to generate a less variable learning target, addressing both intra- and inter-group assignment variance. By reducing intra-group assignment variance, we lower the error term $\text{WVG}(\tilde{\tau})$, and by reducing inter-group assignment variance, we lower the error term $\text{SDMG}(\tilde{\tau})$.

The process begins by partitioning the training dataset into $K$ folds. We then train $K$ primary CATE estimators, each trained on $K - 1$ folds, leaving one fold out. In each primary estimator, the left-out fold is excluded from the training process and is used solely for generating out-of-sample predictions later. After training the primary estimators, for each instance $X_i$, we derive its out-of-sample prediction using the estimator that did not train on $X_i$, with the prediction denoted as $\hat{\tau}_-(X_i)$. Finally, we use the new dataset $\mathcal{D}' = \{(X_i, \hat{\tau}_-(X_i))\}$ to train a regression model as the secondary CATE estimator. Figure 2 provides an overview of CLAGA with $K = 2$. The detailed pseudocode is provided in Appendix B.

## 6.3 INSIGHT

CLAGA introduces a new learning target, $\tilde{\tau}'$, which is the out-of-sample prediction from the primary CATE estimators, $\hat{\tau}_-$. Since $\hat{\tau}_-$ (and consequently $\tilde{\tau}'$) does not depend on the group assignment, we have $\mathbb{E}_\epsilon[\tilde{\tau}'^{(0)}] = \mathbb{E}_\epsilon[\tilde{\tau}'^{(1)}]$, meaning that $\text{SDMG}(\tilde{\tau}') = 0$. Additionally, since the learning targets for both groups become indistinguishable, $\text{WVG}(\tilde{\tau}')$ reduces to $\text{Var}_\epsilon[\tilde{\tau}']$. To further minimize $\text{Var}_\epsilon[\tilde{\tau}']$, various machine learning techniques can be applied to make the primary estimators' predictions more consistent, for example, ensemble learning for primary CATE estimators.

*Remark.* With CLAGA, $\text{SDMG}(\tilde{\tau})$ becomes zero.

*Remark.* With CLAGA, $\text{WVG}(\tilde{\tau})$ can be further minimized using ensemble predictions from the primary estimators.

Lastly, we analyze the error term $(\mathbb{E}_{W,\epsilon}[\tau - \tilde{\tau}^{(W)}])^2$. Since the primary estimators' predictions are used as secondary learning targets, indicating $\mathbb{E}[\tilde{\tau}'] = \mathbb{E}[\hat{\tau}]$, then if these primary estimators are unbiased (i.e. $\mathbb{E}[\hat{\tau}] = \mathbb{E}[\tilde{\tau}]$), it follows that $\mathbb{E}[\tilde{\tau}'] = \mathbb{E}[\hat{\tau}] = \mathbb{E}[\tilde{\tau}]$. Therefore, the error term $(\mathbb{E}_{W,\epsilon}[\tau - \tilde{\tau}^{(W)}])^2$ remains unchanged, meaning CLAGA does not increase or decrease this particular error component.

*Remark.* With CLAGA, the error term $(\mathbb{E}_{W,\epsilon}[\tau - \tilde{\tau}^{(W)}])^2$ remains unchanged if unbiased primary estimators are used.

Table 1: PEHE ratios for each algorithm after applying CLAGA.

| Algorithm | ACIC-2016 | ACIC-2018 |
|---|---|---|
| Single-model | $0.9797_{\pm 0.0610}$ | $0.6036_{\pm 0.3043}$ |
| Two-model | $0.8907_{\pm 0.0941}$ | $0.5009_{\pm 0.2862}$ |
| X-learner | $1.0034_{\pm 0.0692}$ | $0.6277_{\pm 0.2550}$ |
| R-learner | $0.8023_{\pm 0.0973}$ | $0.4622_{\pm 0.2575}$ |
| DR-learner | $0.9340_{\pm 0.0822}$ | $0.4919_{\pm 0.2586}$ |
| TarNet | $0.9189_{\pm 0.1258}$ | $0.5550_{\pm 0.2923}$ |
| CFRNet | $0.9032_{\pm 0.1507}$ | $0.5408_{\pm 0.3073}$ |
| DragonNet | $0.9055_{\pm 0.1502}$ | $0.7090_{\pm 0.2879}$ |
| Avg. PEHE Reduction | 8.3% | 43.9% |

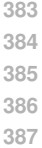
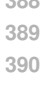
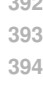
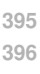
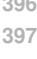
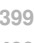

(a) Tree-based models  (b) NN-based models

Figure 3: PEHE across data sizes and model complexities with and without CLAGA.

In summary, our proposed method effectively reduces the error components unrelated to the model training process. When combined with proper model training, CLAGA benefits the performance of CATE estimation.

# 7 EXPERIMENT

In this section, we evaluate the effectiveness of CLAGA. Section 7.1 presents the evaluation of multiple algorithms with and without CLAGA on synthetic datasets. In Section 7.2, we further assess the benefit of CLAGA on real-world datasets.

## 7.1 CLAGA ON SYNTHETIC DATASETS

**Dataset** We evaluate the PEHE of various algorithms, both with and without CLAGA, on two public synthetic datasets: ACIC-2016 (Dorie et al., 2019) and ACIC-2018 (Shimoni et al., 2018). The ACIC-2016 dataset contains 4802 instances with both potential outcomes simulated under 77 different conditions, while ACIC-2018 consists of 24 datasets with sizes ranging between 1000 and 50,000. Additionally, we conduct experiments on the Zenodo dataset (Zhao, 2022), as discussed in Section 4, to examine the effect of CLAGA with respect to data size and model complexity. Further details of the experimental setup, including model implementations and hyperparameters, are provided in Appendix C.

**Results** Table 1 shows the PEHE ratio for each algorithm after applying CLAGA, relative to the original setting (without CLAGA). A ratio of less than 1 indicates improved performance, meaning the application of CLAGA reduced PEHE. The results are averaged across all simulation conditions for each dataset. We observe that CLAGA yields substantial improvements in most settings for both ACIC-2016 and ACIC-2018.

Figure 3 shows the impact of CLAGA on PEHE across different data sizes and model complexities on the Zenodo dataset. In subfigures (a) and (c), CLAGA consistently improves performance across varying data sizes, as indicated by lower PEHE values compared to the baseline. In subfigures (b) and (d), we observe that CLAGA is especially beneficial when model complexity increases, where models are more prone to overfitting. However, in (b) we also find that CLAGA only begins to

Table 2: AUUC results for each algorithm, with and without CLAGA, on four real-world datasets. "Avg. Imp." represents the average improvement of applying CLAGA in each dataset.

| Algorithm | x5 | | Lenta | | Criteo-visit | | Criteo-cv | |
|---|---|---|---|---|---|---|---|---|
| | Vanilla | CLAGA | Vanilla | CLAGA | Vanilla | CLAGA | Vanilla | CLAGA |
| Single | $0.0505 \pm 0.0015$ | **0.0540**$\pm 0.0011$ | **0.0135**$\pm 0.0007$ | $0.0132 \pm 0.0007$ | $0.0302 \pm 0.0001$ | ***0.0309**$\pm 0.0001$ | $0.00308 \pm 0.00004$ | **0.00322**$\pm 0.00009$ |
| Two | $0.0537 \pm 0.0011$ | ***0.0542**$\pm 0.0008$ | $0.0104 \pm 0.0006$ | **0.0128**$\pm 0.0005$ | $0.0245 \pm 0.0001$ | **0.0291**$\pm 0.0000$ | $0.00321 \pm 0.00007$ | **0.00361**$\pm 0.00005$ |
| X-learner | **0.0539**$\pm 0.0005$ | $0.0521 \pm 0.0010$ | $0.0120 \pm 0.0005$ | ***0.0150**$\pm 0.0006$ | $0.0250 \pm 0.0001$ | **0.0284**$\pm 0.0001$ | $0.00295 \pm 0.00009$ | **0.00331**$\pm 0.00006$ |
| R-learner | $0.0428 \pm 0.0012$ | **0.0504**$\pm 0.0010$ | $0.0115 \pm 0.0006$ | **0.0118**$\pm 0.0007$ | $0.0236 \pm 0.0002$ | **0.0277**$\pm 0.0001$ | $0.00287 \pm 0.00009$ | **0.00294**$\pm 0.00008$ |
| DR-learner | $0.0388 \pm 0.0012$ | **0.0410**$\pm 0.0010$ | $0.0121 \pm 0.0004$ | **0.0122**$\pm 0.0005$ | $0.0234 \pm 0.0002$ | **0.0267**$\pm 0.0001$ | $0.00267 \pm 0.00008$ | **0.00303**$\pm 0.00005$ |
| TNet | **0.0414**$\pm 0.0023$ | $0.0402 \pm 0.0048$ | $0.0128 \pm 0.0021$ | **0.0137**$\pm 0.0017$ | $0.0298 \pm 0.0010$ | **0.0304**$\pm 0.0002$ | $0.00372 \pm 0.00020$ | **0.00386**$\pm 0.00005$ |
| CNet | **0.0424**$\pm 0.0025$ | $0.0398 \pm 0.0039$ | $0.0114 \pm 0.0014$ | **0.0119**$\pm 0.0018$ | $0.0291 \pm 0.0005$ | **0.0303**$\pm 0.0003$ | $0.00358 \pm 0.00005$ | ***0.00390**$\pm 0.00004$ |
| DragonNet | $0.0405 \pm 0.0033$ | **0.0424**$\pm 0.0029$ | **0.0115**$\pm 0.0024$ | $0.0108 \pm 0.0014$ | $0.0304 \pm 0.0008$ | **0.0308**$\pm 0.0003$ | $0.00343 \pm 0.00024$ | **0.00365**$\pm 0.00002$ |
| Avg. Imp. | +2.7% | | +6.3% | | +9.0% | | +8.0% | |

provide benefits beyond a certain level of model complexity. We suggest that for models that are too weak, the instability of $\tilde{\tau}'$ may lead to negative effects when applying CLAGA. In summary, CLAGA reduces PEHE when applied with appropriate model training settings, highlighting its effectiveness, especially in scenarios with limited data or higher model complexity.

## 7.2 CLAGA ON REAL-WORLD DATASETS

**Datasets** We evaluate CLAGA on three real-world datasets: x5 (X5-Retail-Group, 2019), Lenta (Lenta, 2020), and Criteo (Diemert Eustache, Betlei Artem et al., 2018). In each dataset, the treatment corresponds to either an advertisement or a communication sent to a customer, and the response is measured as either a user visit or a purchase. The Criteo dataset includes two types of responses, so we treat it as two separate versions: Criteo-cv and Criteo-visit, where the response is either user conversion or user visit, respectively. Further details of the experimental setup, including hyperparameter choices for tree-based and neural network estimators, are provided in Appendix C.

**Evaluation Metric** Since real-world datasets lack ground truth for both potential outcomes, we use the area under the uplift curve (AUUC) (Rzepakowski & Jaroszewicz, 2010) as our evaluation metric, which measures the ranking performance of treatment effects across instances. AUUC integrates the uplift curve, which plots cumulative uplift against the cumulative population targeted by an intervention. In practice, AUUC is often preferred in real-world applications, as practitioners are typically more interested in the relative ranking of treatment effects than the absolute treatment effect values. A higher AUUC indicates that the model successfully ranks instances with higher treatment effects.

**Results** Table 2 presents the AUUC for each algorithm, both with and without CLAGA, on the four real-world datasets. The average improvement in AUUC across all algorithms for each dataset is reported at the bottom. In each pairwise comparison, the algorithm with the better AUUC is highlighted in bold, and the best AUUC for each dataset is marked with an asterisk. Overall, CLAGA demonstrates significant positive improvements in most cases. The best performance is observed when CLAGA is applied across all datasets.

## 8 CONCLUSION

In this paper, we identified an overlooked issue in CATE estimation: the inconsistency of learning targets across treatment groups, which contributes directly to prediction error. To address this, we proposed CLAGA, a novel approach designed to reduce variance-related errors by enforcing consistent labeling across group assignments. Comprehensive experiments on both synthetic and real-world datasets demonstrate that CLAGA improves the performance of a wide range of CATE estimation algorithms, particularly in settings prone to overfitting.

Beyond empirical gains, our error decomposition framework provides new insights into the sources of error in CATE estimation, clarifying how variance reduction can yield more accurate and reliable treatment effect estimates. By tackling these variance-related challenges, CLAGA opens up new directions for improving estimator robustness and advancing reliable personalized decision-making. A more detailed discussion of the limitations of our approach is provided in Appendix D.

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

## A  ERROR DECOMPOSITION DERIVATION

In this appendix, we provide the full derivation of the error decomposition discussed in Section 5. We begin with the precision in estimating heterogeneous effects (PEHE), defined as:

$$\mathbb{E}_{M,W,\epsilon}[(\hat{\tau}(x) - \tau(x))^2]. \tag{5}$$

Here, $\hat{\tau}(x)$ is the model prediction, $\tau(x)$ is the ground truth of treatment effect, $M$ denotes randomness from model training (e.g., initialization or optimization), and $W$ is the treatment assignment. $\epsilon$ captures the residual variation in the observed outcomes, which may arise either from measurement error or from inherent stochasticity of the outcome distribution (e.g., Bernoulli variability). Specifically, the observed potential outcomes are assumed to follow

$$Y(w) = \mu_w(X) + \epsilon_w,$$

where $\mathbb{E}[\epsilon_w \mid X] = 0$.

For notational simplicity, we drop the dependency on $x$ and write $\hat{\tau}$, $\tau$, and $\tilde{\tau}$ instead of $\hat{\tau}(x)$, $\tau(x)$, and $\tilde{\tau}(x)$, respectively.

Recall that $\tilde{\tau}^{(w)}$ denotes the group-specific learning target under treatment assignment $w \in \{0,1\}$, which may depend on the observed outcome and thus on $\epsilon_w$. We then define

$$\tilde{\tau} = \begin{cases} \tilde{\tau}^{(0)}, & \text{if } W = 0 \\ \tilde{\tau}^{(1)}, & \text{if } W = 1. \end{cases}$$

STEP 1: DECOMPOSITION OF PEHE

We begin by introducing the learning target $\tilde{\tau}$, and expanding PEHE as follow:

$$\begin{aligned} \mathbb{E}_{M,W,\epsilon}[(\hat{\tau} - \tau)^2] &= \mathbb{E}_{M,W,\epsilon}[((\hat{\tau} - \tilde{\tau}) + (\tilde{\tau} - \tau))^2] \\ &= \mathbb{E}_{M,W,\epsilon}[(\hat{\tau} - \tilde{\tau})^2] \\ &\quad + \mathbb{E}_{M,W,\epsilon}[(\tau - \tilde{\tau})^2] \\ &\quad - 2\mathbb{E}_{M,W,\epsilon}[(\hat{\tau} - \tilde{\tau})(\tau - \tilde{\tau})]. \end{aligned} \tag{6}$$

Note that the second term depends only on $W$ and $\epsilon$ and not on model randomness $M$, and can be rewritten as:

$$\mathbb{E}_{W,\epsilon}[(\tau - \tilde{\tau})^2].$$

STEP 2: DECOMPOSITION OF $\mathbb{E}_{W,\epsilon}[(\tau - \tilde{\tau})^2]$

Focusing on the second term, we expand using the variance identity:

$$\mathbb{E}_{W,\epsilon}[(\tau - \tilde{\tau})^2] = \text{Var}_{W,\epsilon}[\tilde{\tau}] + \left(\mathbb{E}_{W,\epsilon}[\tau - \tilde{\tau}]\right)^2. \tag{7}$$

Since $\tau$ is deterministic given $x$, we have $\text{Var}[\tau] = 0$ and $\text{Cov}[\tau, \tilde{\tau}] = 0$. Thus no cross-term with $\tau$ appears.

STEP 3: VARIANCE OF LEARNING TARGETS

As $\tilde{\tau}$ depends on both $W$ and outcome noise $\epsilon$, the variance is

$$\begin{aligned} \text{Var}_{W,\epsilon}[\tilde{\tau}] &= (1 - \pi) \text{Var}_\epsilon[\tilde{\tau}^{(0)}] + \pi \text{Var}_\epsilon[\tilde{\tau}^{(1)}] \\ &\quad + \pi(1 - \pi)\left(\mathbb{E}_\epsilon[\tilde{\tau}^{(0)}] - \mathbb{E}_\epsilon[\tilde{\tau}^{(1)}]\right)^2, \end{aligned} \tag{8}$$

where $\pi = \Pr(W = 1|x)$.

STEP 4: FINAL DECOMPOSITION

Combining Equations 6, 7, and 8, we obtain:

$$\mathbb{E}_{M,W,\epsilon}[(\hat{\tau} - \tau)^2] = \underbrace{\mathbb{E}_{M,W,\epsilon}[(\hat{\tau} - \tilde{\tau}^{(W)})^2]}_{\text{Model Error}} - \underbrace{2\mathbb{E}_{M,W,\epsilon}[(\tau - \tilde{\tau}^{(W)})(\hat{\tau} - \tilde{\tau}^{(W)})]}_{\text{Model-Target Covariance}}$$

$$+ \underbrace{(1 - \pi)\text{Var}_\epsilon[\tilde{\tau}^{(0)}] + \pi\text{Var}_\epsilon[\tilde{\tau}^{(1)}]}_{\text{Group Assignment Weighted Variance}} + \underbrace{\pi(1 - \pi)(\mathbb{E}_\epsilon[\tilde{\tau}^{(0)}] - \mathbb{E}_\epsilon[\tilde{\tau}^{(1)}])^2}_{\text{Inconsistency Across Group Assignments}} \quad (9)$$

$$+ \underbrace{(\mathbb{E}_{W,\epsilon}[\tau - \tilde{\tau}^{(W)}])^2}_{\text{Bias of Learning Target}},$$

This aligns directly with the error terms in Section 5, showing how inconsistency in learning target construction contributes directly to the final estimation error.

Each term reflects a distinct source of error:

- **Model Error:** The expected squared error between model prediction $\hat{\tau}$ and the learning target $\tilde{\tau}$, capturing the model's inability to fit its assigned target.

- **Model-Target Covariance:** The interaction between the bias of learning target $(\tau - \tilde{\tau})$ and model error $(\hat{\tau} - \tilde{\tau})$. A negative value suggests the model compensates for the bias of learning target, partially correcting toward the true $\tau$.

- **Group Assignment Weighted Variance:** The intra-group variance of learning targets $\tilde{\tau}^{(0)}$ and $\tilde{\tau}^{(1)}$, weighted by group assignment probabilities. High within-group variance leads to uncertainty in supervision signals.

- **Inconsistency Across Group Assignments:** The squared difference between the group-conditional mean learning targets. This reflects how differently an estimator learns for the same instance under different group assignments.

- **Bias of Learning Target:** The average squared difference between learning target $\tilde{\tau}$ and the true effect $\tau$, reflecting structural bias induced by the algorithm's label construction design.

# B  CLAGA ALGORITHM

---

**Algorithm 1:** Consistent Labeling Across Group Assignments (CLAGA)

---

**Data:** Training dataset $\mathcal{D} = \{(X_i, W_i, Y_i)\}$, CATE estimation algorithm **G**, number of folds $K$
**Result:** CATE estimator $\mathbf{g}'$ using CLAGA
`// Step 1: Partition the dataset`
Partition $\mathcal{D}$ into $K$ folds: $\mathcal{D}_1, \mathcal{D}_2, \ldots, \mathcal{D}_K$
`// Step 2: Train primary estimators on` $K-1$ `folds`
**for** $i = 1$ *to* $K$ **do**
    Train primary estimator $\mathbf{g}_i$ using $\mathcal{D} - \mathcal{D}_i$ (i.e., all folds except $\mathcal{D}_i$)
    Generate out-of-sample predictions $\hat{\tau}_-(X_j)$ for each instance $X_j \in \mathcal{D}_i$
**end**
`// Step 3: Create relabeled dataset`
Create new dataset $\mathcal{D}' = \{(X_i, \hat{\tau}_-(X_i))\}$, where $\hat{\tau}_-(X_i)$ is the out-of-sample prediction from the primary estimator
`// Step 4: Train secondary estimator`
Train a regression model $\mathbf{g}'$ on the entire relabeled dataset $\mathcal{D}'$
**return** Final CATE estimator $\mathbf{g}'$

---

## C    EXPERIMENTAL SETUP

### CLAGA ON SYNTHETIC DATASETS

We use eight CATE estimation algorithms as mentioned in Section 3.1. For each algorithm, we train models both with and without CLAGA and evaluate their PEHE on test data. The algorithms are implemented using the Python package CausalML (Chen et al., 2020), with LGBMClassifier and LGBMRegressor (Ke et al., 2017) for all tree-based estimators. For neural network (NN) estimators (TarNet, CFRNet, DragonNet), we follow the architectures described in their original papers and implement them in PyTorch.

For ACIC-2016 and ACIC-2018, due to the relatively small data sizes, we use 'n_estimators=100' and 'num_leaves=32' for tree-based models. For the larger Zenodo dataset, we use 'n_estimators=400' and 'num_leaves=64'. A subsample rate of 0.5 and 'subsample_freq=3' are applied across all datasets, with other hyperparameters left as default. In CLAGA, we use $K = 10$ for ACIC-2016 and ACIC-2018, and $K = 2$ for Zenodo. Each experiment setting is repeated 10 times with different random seeds.

For NN estimators, we adopt the following configurations. On ACIC-2016 and ACIC-2018, TarNet and CFRNet use shared layers of (100, 100), head layers of (50, 50), 100 training epochs, learning rate $1 \times 10^{-3}$, and batch size 64. DragonNet uses shared layers of (100, 100, 80, 60), head layers of (60, 40, 20, 10), with the same epoch, learning rate, and batch size settings. On Zenodo, TarNet and CFRNet use shared layers of (200, 200), head layers of (100, 100), 200 training epochs, learning rate $1 \times 10^{-3}$, and batch size 64. DragonNet uses shared layers of (200, 200, 160, 120), head layers of (80, 60, 40, 20), again trained for 200 epochs with learning rate $1 \times 10^{-3}$ and batch size 64.

### CLAGA ON REAL-WORLD DATASETS

The general setup follows that of Section 7.1. For x5 and Lenta, we use 'n_estimators=100' and 'num_leaves=64', while for Criteo datasets, we use 'n_estimators=500' and 'num_leaves=256', reflecting the larger dataset size. For CLAGA, we set $K = 2$ for all datasets.

For neural network estimators, we use the same hyperparameter settings as in the synthetic experiments: - For x5 and Lenta, we adopt the same configurations as in ACIC-2016/2018. - For Criteo-cv and Criteo-visit, we adopt the same configurations as in Zenodo.

## D    LIMITATIONS

While CLAGA has shown promising results in reducing variance-related errors and improving CATE estimation, there are several limitations to our approach that merit consideration.

**Dependence on the base CATE estimators**    The effectiveness of CLAGA is closely tied to the primary CATE estimators. Although CLAGA reduces variance-related errors, it relies on the quality of the initial estimators. If these estimators exhibit significant bias or are overly simplified, the overall improvement may be limited. Thus, in scenarios where the primary estimators are weak, CLAGA may not fully realize its potential to enhance CATE estimation.

**Increased computational complexity**    The process of training multiple base estimators during the K-fold procedure can significantly increase the overall training time and computational resources, particularly for large datasets or highly complex models. This added computational complexity may limit the practicality of CLAGA in resource-constrained environments where efficiency is a concern.

**Theoretical analysis to PEHE**    Our theoretical analysis and error decomposition are specifically focused on minimizing the PEHE as the evaluation metric. While PEHE is widely used in CATE estimation, the error decomposition framework we propose may not directly be guaranteed to extend to other metrics, such as AUUC.

## E  THE USE OF LARGE LANGUAGE MODELS

In preparing this work, we made use of large language models (LLMs) as a general-purpose assist tool. Specifically, LLMs were used to:

- provide writing assistance, including language polishing, rephrasing for conciseness, and suggesting structural improvements to paragraphs;
- check grammar, style consistency, and readability of the manuscript;
- offer suggestions for figure captions and formatting in LaTeX;
- assist in identifying relevant references based on author-provided keywords, with all references verified and cross-checked by the authors.

LLMs were not used to generate research ideas, conduct experiments, derive results, or perform data analysis. All scientific contributions, methodology, derivations, and experimental designs are the sole work of the authors. The authors take full responsibility for the contents of this paper.

