# OpenReview forum: "Consistent Labeling Across Group Assignments: Variance Reduction in Conditional Average Treatment Effect Estimation"
_ICLR.cc/2026/Conference — ICLR 2026 Conference Withdrawn Submission_

### Official Review · Reviewer_Xq6g · 2025-10-25

**Soundness:** 1
**Presentation:** 3
**Contribution:** 1
**Rating:** 2
**Confidence:** 4

**Summary:**

The paper introduces a new framework built on top of existing CATE estimation methods to address what the authors describe as an inconsistency in how “vanilla” estimators perform across different treatment assignment groups. This inconsistency is quantified using a proposed metric called the Discrepancy Ratio (DR). The framework, termed Consistent Labeling Across Group Assignments (CLAGA), aims to mitigate this claimed inconsistency.

**Strengths:**

- The authors clearly demonstrate a problem of cate estimation
- The authors provide theoretical insight into the estimation error via decomposition, which may be a useful tool for discussion and analysis
- The authors present a new framework intended to address the identified inconsistency in CATE estimation

**Weaknesses:**

- By developing their own metric on which they showcase a problem and own solution, the authors develop their own niche with no immediate comparison work available. Nevertheless other works concerned with overall CATE estimators performance exist and should be included. For example, algorithmic fairness research may be directly relevant here.
- The presented issue and consequent solution are motivated by Figure 1, which is correctly caption as showcasing overfitting. Method for tackling overfitting are ubiquitous across machine learning disciplines. Discussion and comparison to e.g. methods with regularization or attempts at showing double descent seem like natural follow ups, but they are not discussed here
- The DR learner and R learner have been shown to be asymptotically optimal. From that perspective nothing better can be done and only finite sample performance can be improved. Discussion of asymptotic point of view is missing.
- The proposed method itself further splits the data on top of already present cross fitting necessary by existing methods such as the DR learning. This leads to further decrease of dataset size used to fit each stage. This disadvantage seems significant and is not discussed.
- The proposed method introduces a new supervised learning step with synthetic labels on top of existing works. Such approach can be intuitively seen simply as a form of regularization for overfitting. Since the additional layer does not train wrt targets from the data but synthetic ones, no new information is being extracted from the dataset - only regularization occurs. In this sense, the method closely resembles stacking or self-distillation. This interpretation suggests that CLAGA’s gains may stem from known effects of cross-fitting and stacking, yet this crucial perspective is not discussed in the paper.
- Viewing the method as a form of regularization to overfitting, the proposed approach seems excessively expensive.
- The authors state that CLAGA “does not rely on unconfoundedness” (lines 113–114), yet CATE estimation fundamentally does. Since CLAGA is presented as a framework for CATE estimation, this statement is misleading.
- Code is not released, restricting reproducibility.

**Questions:**

- How does the proposed problem relate to work in fairness research which also concerns itself with balancing estimation error across groups? How is it different?
- If the identified problem is overfitting, how do other regularization strategies that also reduce overfitting at much lower cost perform?
- How do the authors view their solution wrt asymptotic vs finite sample statistics point of view?
- Considering the added step regresses towards synthetic labels - how do the authors view their method from the information theoretic point of view which would imply no new information is being extracted from the data?
- How does the framework perform on newer baselines?

---

### Official Review · Reviewer_Af5n · 2025-10-31

**Soundness:** 2
**Presentation:** 1
**Contribution:** 1
**Rating:** 2
**Confidence:** 3

**Summary:**

This work proposes a metric for evaluating CATE estimators.

Despite many years of experience in developing CATE estimators, I couldn't make sense of the property the metric is attempting to evaluate (line 191) or the justification given in Section 5.1. Perhaps there's something fundamental that I'm missing. I'm very curious to hear what the other reviewers thought and am looking forward to engaging with the authors during the rebuttal period to see if there's something that I'm missing.

**Strengths:**

Its merits aside, the proposed metric seems to be novel.

**Weaknesses:**

I don't understand what line 191 means. If you condition on $X=x$, then the value of $W$ doesn't matter. So I understand line 191 to mean that you're taking an expectation with respect to the training data, and only the training data.

The proposed discrepancy measure seems something about the overfitting of the estimators under a strong null hypothesis of both (i) treatment randomized independently of covariates and (ii) no distributional treatment effect. I don't understand why this setting would be of interest, as it (i) imposes conditions beyond those required of most CATE estimators and (ii) imposes conditions on higher-order moments of the counterfactual distributions that have nothing to do with the CATE.

**Questions:**

Minor comment on references on line 47:
* R-learner was first proposed in Corollary~9.1 of
Robins, James M. "Optimal structural nested models for optimal sequential decisions." Proceedings of the Second Seattle Symposium in Biostatistics: analysis of correlated data. New York, NY: Springer New York, 2004.
* DR-learner was proposed in Section 3.1 of
van der Laan, Mark J. "Targeted Learning of an Optimal Dynamic Treatment, and Statistical Inference for its Mean Outcome." (2013).

---

### Official Review · Reviewer_PipY · 2025-11-03

**Soundness:** 2
**Presentation:** 1
**Contribution:** 1
**Rating:** 2
**Confidence:** 4

**Summary:**

The paper considers an issue in CATE estimation: many algorithms exhibit inconsistent learning behavior for the same instance across different group assignments.
Next, the paper argues that the inconsistency indeed contributes to higher test errors and cannot be resolved through conventional machine learning techniques. To address this problem, the paper proposes a new algorithm.

**Strengths:**

The paper provides simulation and real-world studies.

**Weaknesses:**

1. The paper's theoretical foundation is excessively simple, with key arguments' validity contingent upon vague definitions and assumptions. If the authors claim "theoretical analysis", they must rigorously align with established literature in the field. For instance, the paper's use of "consistency" appears misapplied.

2. The manuscript seems preoccupied with the issue of unbiased estimators. It is overlooked that even biased estimators may converge to the true parameter as sample size increases. Given large samples prevalent in contemporary applications, large-sample theory would potentially justify such estimators.

3. Extensive literature on causal inference using machine learning methods has demonstrated that naive applications invariably fail, whereas strategic modifications can yield suitable estimators. In that, the finding in the paper is not new. The existing studies have substantially gone beyond primitive mean-variance analysis, rendering the paper's simplistic derivations seemingly disconnected from these methodological advances.

**Questions:**

Please find the items in Weaknesses.

---

### Official Review · Reviewer_2K6c · 2025-11-04

**Soundness:** 3
**Presentation:** 2
**Contribution:** 3
**Rating:** 4
**Confidence:** 4

**Summary:**

The paper proposes new objective when estimating CATE to solve the problem of inconsistent learning behavior for different treatment groups. The paper proposes a decomposition of PEHE to motivate the method and proposes a new algorithm. The paper concludes with experiments showing the effectiveness of the algorithm.

**Strengths:**

Originality: I like the idea and I think the decomposition and the extra layer of thinking on estimating CATE with PEHE as loss is quite original.

Quality: The paper has good theoretical justification, comprehensive experiments and clear algorithm descriptions.

Significance: I believe this consistency among group property is quite important in some applications.

**Weaknesses:**

I like the paper but I think the main weakness is the clarity of the presentation and the experiments. Details below:

Clarity:
1. line 191: I found the notation quite confusing. $\hat{\tau}(x)$ is usually a deterministic quantity with lower case $x$.
2. Similar for line 237, the notation is quite confusing to me.
3. Line 323: this is quite confusing, what is well-trained and what is well-designed.
4. line 372 to 3745: still quite confusing. In particular I don't see how these different $\tau$'s come up. I get what you are trying to convey here though.


Experiments:
1. The discrepancy ratio is cool but isn't that testing tons of hypothesis? Maybe you should use a lower p-value.
2. How do you calculate PEHE in your experiments? This is quite trivial but I think at least should be mentioned.
3. In table 1, why do the numbers in 2016 >> 2018?
4. Also in table 1, I don't quite get why you use the ratio as a metric to compare before and after.
5. Figure 3 does not have sd bars.
6. In Table 2, why the first dataset is way worse?

**Questions:**

1. For figure 1(c): your method reduces errors unrelated to the training process, which basically says everything else the same, using CLAGA should be better than say using DR-learner. But why there is a much smaller almost horizontal line for CLAGA in figure 1, which from figure 3 is not necessarily true? I guess my confusion comes from how is this method solving the problem of figure 1(b) and (c).

---

### Note · Authors · 2025-11-16

I have read and agree with the venue's withdrawal policy on behalf of myself and my co-authors.